# Recent Progress of Polymeric Corrosion Inhibitor: Structure and Application

**DOI:** 10.3390/ma16082954

**Published:** 2023-04-07

**Authors:** Xuanyi Wang, Shuang Liu, Jing Yan, Junping Zhang, Qiuyu Zhang, Yi Yan

**Affiliations:** 1Chongqing Technology Innovation Center, Northwestern Polytechnical University, Chongqing 401135, China; 2Department of Chemistry, School of Chemistry and Chemical Engineering, Key Laboratory of Special Functional and Smart Polymer Materials of Ministry of Industry and Information Technology, Northwestern Polytechnical University, Xi’an 710129, China

**Keywords:** metal corrosion, anti-corrosion inhibitor, polymeric inhibitor, adsorption

## Abstract

An anti-corrosion inhibitor is one of the most useful methods to prevent metal corrosion toward different media. In comparison with small molecular inhibitors, a polymeric inhibitor can integrate more adsorption groups and generate a synergetic effect, which has been widely used in industry and become a hot topic in academic research. Generally, both natural polymer-based inhibitors and synthetic polymeric inhibitors have been developed. Herein, we summarize the recent progress of polymeric inhibitors during the last decade, especially the structure design and application of synthetic polymeric inhibitor and related hybrid/composite.

## 1. Introduction

Metal corrosion has become a global problem, which not only induce accident due to the decrease in the mechanical strength, but also causes huge economic losses. Corrosion inhibitor is one of the most useful and economic strategies to protect metal materials toward different media. Generally, there are several kinds of inhibitors, including inorganic inhibitors, organic inhibitors, and polymeric inhibitors [1,2,3]. Compared with inorganic inhibitors, organic inhibitors and polymeric inhibitors are much cheaper and more powerful. More importantly, both organic inhibitors and polymeric inhibitors can be rationally designed and easily synthesized. It is well-known that the adsorption of inhibitors on the surface of metal and corresponding adhesion property performs important role in the application of inhibitors [4]. Therefore, adsorption groups are broadly used in the structure design of inhibitors. Some pioneered review papers have already summarized the progress of organic inhibitors [5,6].

In comparison with small molecular organic inhibitor, polymeric inhibitors show the following advantages (as illustrated in Figure 1): (i) more adsorption groups can be introduced to one molecule by tuning the number of the repeat unit; (ii) different adsorption groups can be integrated into one polymer via copolymerization (for example copolymerization of monomer A and monomer B), which may generate a synergetic adsorption effect; (iii) the configuration of supramolecular self-assembly of polymeric inhibitor allows the structure optimization of polymeric inhibitor to achieve best adsorption performance; and (iv) the flexibility and mobility of the polymer chain provides processability, which also allow the formation of hybrid/composite with inorganic inhibitor to achieve improved anti-corrosion performance.

Heterocyclic compounds (as shown in Figure 1) have been considered to be excellent corrosion inhibitors, owing to the dense electron centers of heteroatom; however, their synthetic process is usually very harmful to the environment. The adsorption sites of polymeric inhibitor can be increased by increasing their molecular weight (in other words, the number of repeat unit) and can be a potential candidate for the use of heterocyclic compounds in corrosion protection [7]. The mechanism by which polymers can perform corrosion inhibition is the metal-polymer bonding mechanism. Polymeric corrosion inhibitors dissolved in corrosive electrolyte can be adsorbed on the metal surface to form polymeric film through the metal-polymer bonding mechanism, as shown in Figure 2. The adsorption mechanism of polymers containing heteroatoms discussed in this mini-review mainly follows such mechanisms and forms polymer film on the metal surface through chemisorption or physical-chemical adsorption (mixed adsorption) mechanisms [8,9]. The hydrophilicity and hydrophobicity of the polymer determine whether the polymer film formed on the metal surface can effectively protect against corrosion and whether it can exist stably. A suitable ratio between hydrophilic and hydrophobic segments is essential to the corrosion inhibitive performance of polymeric inhibitor [8,10,11].

According to the source and synthetic method, there are generally two kinds of polymeric inhibitors, natural polymer-based inhibitors [12,13] and synthetic polymeric inhibitors [14,15]. With regarding to the design of polymeric inhibitor, on one hand, the presence of heteroatom is crucial for polymeric corrosion inhibitors and greatly affects the corrosion inhibitive properties, which are currently known to be O < N < S < P according to their capability of coordination [16]. One the other hand, most of the polymeric inhibitors belong to polyelectrolyte, both the charged group and its counterion are important.

Herein, we are trying to summarize the progress on the design and application of polymeric inhibitors during the last decade. Generally, both inhibitors from natural polymer and synthetic polymer and their related hybrid/composite will be covered, which may be helpful to researchers, entrepreneurs, and engineers in the fields of functional polymer and metal protection.

## 2. Corrosion Inhibitor from Natural Polymer

Natural polymers (Figure 3), such as starch, cellulose, chitosan, and other natural polysaccharide substances extracted from natural products, have been historically used as corrosion inhibitor [17] due to the presence of multiple coordination groups in their structures. Generally, the inhibition property of natural polymers depends on the number of their functional groups, which usually varies according to their source. Therefore, different strategies have been developed to extract natural products from high content raw plants, such as solvent extraction, mechanical extrusion, pre-gelatinization treatment, and so on.

Starch-base inhibitors were broadly used due to their abundance in nature. For example, Kaseem et al. found that the addition of starch to the electrolyte can protect Al-Mg-Si alloy via the formation of α-Al_2_O_3_ on the surface, while the oxide layer of the alloy without the starch addition consists of *γ*-Al_2_O_3_ only [18]. Thamer et al. used acid hydrolysis to synthesize starch nanocrystal and investigated its corrosion inhibition for mild steel in 1 M HCl. The maximum inhibition efficiency was 67% when the concentration of the nanocrystal was 0.5 g/L [19]. Physical blend of starch and pectin was developed by Sushmitha and coworkers [20]. They found that such a blend can be used as corrosion inhibitor for mild steel in 1 M HCl. The corrosion inhibition efficiency was up to 74% when the temperature was controlled at 30 °C. Furthermore, the addition of this blend to the epoxy resin can effectively reduce the porosity of the coating and therefore enhance the corrosion inhibition efficiency.

Zhang et al. investigated the corrosion inhibition of steel by maize gluten meal extract in simulated concrete pore solution containing 3% NaCl using electrochemical techniques and showed that the corrosion resistance of steel was improved after the introduction of the extract [21]. The corn protein powder extract is a mixed inhibitor causing an inhibitory effect on both anodic and cathodic reactions, where polar groups, such as N and O in the amide group, perform the main adsorptive effect.

However, the crystalline regions decreased their water solubility and limited corresponding application in metal corrosion protection [22]. Different strategies have been developed to increase the water solubility of starch. For example, *gadong* tuber starch (GTS)-based inhibitor was extracted from *Dioscorea hispida* and dispersed in 90% dimethyl sulfoxide [23]. The corrosion inhibition property was evaluated by using SAE 1045 carbon steel in 0.6 M NaCl, giving inhibition efficiency as high as 86.3% at GTS concentration of 1500 ppm. Extrusion can also be used to physically modify starch by increasing the amorphous region. Anyiam et al. [24] used an alkaline treatment with NaOH to modify starch extracted from sweet potato and investigated its corrosion inhibitor performance on mild steel at 0.25 M H_2_SO_4_ by gravimetric and potentiodynamic polarization (PDP) techniques. The alkaline treated starch can achieve inhibitive efficiency as high as 84.2%, which was found to be more effective than the unmodified starch in corrosion inhibition. Meanwhile the addition of KI could perform a synergistic role to enhance the inhibition effect. They also modified starch by extrusion and found that the modified starch can give inhibition efficiency up to 64% at concentration as low as 0.7 g/L in 1 M HCl solution [25].

To improve the water solubility of natural starch, Deng et al. prepared cassava starch-acrylamide graft copolymer (CS-AAGC) by grafting cassava starch with acrylamide using (NH_4_)_2_S_2_O_8_ and NaHSO_3_ as initiators [26]. Weight loss and electrochemical analysis revealed that CS-AAGC is a hybrid inhibitor for aluminum in 1 M H_3_PO_4_, which performs a major role in inhibiting the anodic reaction with a maximum inhibition efficiency of 90.6%.

Grafting starch with other kinds of water-soluble monomers can also be used to prepare starch-based inhibitors. For example, Hou et al. [27] modified starch by grafting with acrylic acid (AA) and investigated the corrosion inhibition performance on Q235 carbon steel under HCl environment by weight loss and electrochemistry. It was found that 200 ppm of such terpolymer can give inhibitive efficiency up to 90.1% at 30 °C. Acrylamide is generally used as a co-monomer in the modification of starch. For example, Li et al. [28] grafted acrylamide onto cassava starch and studied the corrosion inhibition effect on aluminum in 1 M HNO_3_. According to the results of the weight loss and electrochemical study, the acrylamide-grafted starch showed better anti-corrosion performance than the un-grafted one. While in a similar study, Wang et al. [29] found that the acrylamide (AM)-grafted starch can also be used as inhibitor for the protection of zinc in 1 M HCl with inhibition efficiency as high as 92.2%. Furthermore, they grafted sodium allyl sulfonate and acrylamide onto tapioca starch [30]. It was found that such terpolymer excellent anti-corrosion property on cold-rolled steel, the inhibition efficiency can be achieved as high as 97.2% for 1 M HCl and 90% for 1 M H_2_SO_4_, respectively.

Comparing with starch, cellulose derivatives are also widely used as inhibitors due to their easy modification [31]. Methyl hydroxyethyl cellulose (MHEC) shows good water solubility and facilitates its adsorption on metal surface [32]. Eid et al. [33] investigated the corrosion inhibition effect of MHEC on copper in 1 M HCl solution by cyclic voltammetry, potentiodynamic polarization, and weight loss techniques. It was found that such inhibitors could prevent the mass and charge transfer on the copper surface by calculating the activation energy (E_a_) and heat of adsorption (Q_ads_) of the adsorption process. The corrosion efficiency can achieve as high as 90% at concentration as low as 4 g/L. In addition to copper, cellulose derivatives can also be used to protect aluminum alloys. Nwanonenyi et al. [34] found that hydroxypropyl cellulose (HPC) was a good inhibitor for aluminum in acidic media (0.5 M HCl and 2 M H_2_SO_4_) at 30–65 °C.

Wang et al. [35] prepared thiocarbonylhydrazide modified glucose derivatives (Figure 3) through *N*-glycosyl linkage at the C-1 of the saccharide moiety and applied them to the corrosion protection of N80 carbon steel pipelines in the oil and gas industry. It was found that these compounds showed excellent corrosion protection with corrosion inhibition efficiencies of 99.1% and 99.4%, respectively.

Different from starch and cellulose, the presence of amino groups in chitosan (CS) provide additional adsorption groups and enables improved inhibition property. For example, Wen and coworkers investigated the effect of water-soluble chitosan corrosion inhibitor on the corrosion behavior of 2205 duplex stainless steel in NaCl and FeCl_3_ solutions [36]. When immersed in 0.2 g/L chitosan solution for ca. 4 h, the surface of stainless steel specimen will be covered by a dense and uniform adsorption film, and the corrosion inhibition efficiency can achieve as high as 62%. In order to improve the corrosion inhibition performance of CS, chemical modification, such as esterification and grafting on chitosan, has been developed [37]. For example, Zhang et al. [38]. introduced extra adsorption groups to CS by using facile Schiff base reaction and etherification. As shown in Figure 4, the synthesized CS-1 and CS-2 exhibited excellent anti-corrosion property on Q235 mild steel in 1 M HCl solution. It was found that the introduction of pyridine groups increased the absorption of inhibitors on the metal surface comparing with crude CS. Moreover, the presence of hydrophobic phenyl ring in CS-2 enhanced the electron density at active sites and gave better surface coverage than CS-1, achieving inhibition efficiency as high as 98.0% at a low concentration of 150 mg/L at 25 °C.

In addition to the above three types of natural polymer-typed inhibitors, other natural polysaccharides can also be used as metal corrosion inhibitors, such as gelatin, pectin, carrageenan, xanthan gum, sodium alginate, and so on. Fares et al. [39] investigated the anti-corrosion property of ι-carrageenan for aluminum in the presence of pefloxacin mesylate and 1.5 M and 2.0 M HCl. It was revealed that pefloxacin acted as zwitterion mediator for the adsorption of carrageenan on the metal surface, and therefore, gave an increase in the inhibition efficiency from 67% in the absence of the medium to 92%. Similarly, pectin is also used as corrosion inhibitor [40]. Grassino et al. [41] extracted pectin from tomato peel waste (TPP) and used it as tin corrosion inhibitor. Compared with the corrosion inhibitor of commercial apple pectin, it was found that TPP mainly performed a cathodic protection role and was an effective corrosion inhibitor for tin in NaCl, acetic acid and citric acid solutions.

To compare the corrosion inhibition property of different natural polymers, Chen et al. investigated the scale and corrosion inhibition properties of polyaspartic acid (PASP), polyepoxysuccinic acid (PESA), oxidized starch (OS), and carboxymethyl cellulose (CMC) by molecular dynamics simulations and density flooding theory calculations, and the binding energies of these substances to Fe surfaces were: PASP > OS > CMC > PESA [42].

Owing to their abundant resources, the conversion of natural polymers to carbon dots (CDs) became an emerging research topic in the design of novel inhibitors. For example, He et al. [43] developed a green and high-yielding CDs-based pickling solution using a range of low-cost and easily available sugars, such as glucose, soluble starch, fructose, or sucrose, and using concentrated H_2_SO_4_ as the oxidant and an acid source for the preparation of CDs. The corrosion inhibition performance of CDs on Q235 carbon steel in 0.5 M H_2_SO_4_ solution was subsequently determined by weight loss test, electrochemical impedance spectroscopy (EIS), and PDP measurement. It was found that the corrosion inhibition of carbon steel could reach about 95% at a low precursor concentration of 0.26%. Based on the electrochemical and corrosion surface analysis, it can be reasonably assumed that the corrosion inhibition and protection effect of CDs is exerted by forming a protective film on the metal surface.

Overall, although natural polymer-based corrosion inhibitors (Table 1) show the advantages of wide source, environmentally friendly, easy to obtain and degrade, their chemical modification and modulation of adsorption groups, and corrosion inhibition efficiency is still more challenging As an alternative strategy, synthetic polymeric corrosion inhibitors provide precision structure and architecture design; therefore, the concept of synthetic polymer-based corrosion inhibitor was also developed.

## 3. Synthetic Polymeric Corrosion Inhibitor

In principle, both coordination group and charged group can be potential adsorption sites for metal surface. To design effective inhibitors, both the adsorption site and the polymeric structure perform important roles. The development of polymer chemistry, especially controlled radical polymerization techniques, provides accurate architecture and structure control over the number of repeat units, functional groups, and the topology. In this section, different kinds of synthetic polymeric inhibitors will be discussed according to their polymer skeletons and adsorption sites, such as phosphorus, sulfur, nitrogen, and so on.

### 3.1. Phosphorus-Containing Synthetic Polymeric Inhibitor

Phosphorus can coordinate with metal surfaces more strongly than oxygen, which has been introduced to polymeric inhibitor as additional adsorption site in the form of both coating and solution. Generally, there are two kinds of strategies to prepare phosphorus-containing synthetic polymeric inhibitor: (i) post-modification of natural polymer [44]; and (ii) direct polymerization of phosphorus-containing monomer [45].

Grafting phosphorus-containing groups to natural polymers not only can introduce additional adsorption sites, but also improve their water solubility. David et al. functionalized chitosan with phosphonic acids (P-1 in Figure 5) via the Kabachnik-Fields reaction and subsequent hydrolysis [44]. The introduction of phosphonic acid groups greatly decreased the dynamic viscosity of chitosan. Moreover, compared with the native chitosan, P-1 showed improved adsorption on the carbon steel surface and better corrosion inhibition property. Meanwhile, such material can also be used in metallic aluminum corrosion protection. David et al. [44] fabricated inhibitive coating by the layer-by-layer (L-b-L) technique from native chitosan or synthesized phosphorylated chitosan (P-1 in Figure 5) combined with alginate functionalized chitosan. It was found that the coating can create a physical barrier that acts mainly by reducing the active surface area and has the effect of blocking the penetration of the aggressive species into the metal substrate. According to the result of electrochemical impedance spectroscopy (EIS) in 0.1 M Na_2_SO_4_ solution, the as prepared coating showed improved corrosion resistance for aluminum alloy 3003.

Phosphonic acids can also be introduced to chitin. As demonstrated by Hebalkar and coworkers, they synthesized phosphorylated chitin (P-2 in Figure 5) to solve the water solubility of native chitin and improve their coordination ability with metal surface [46]. According to the gravimetric analysis in NaCl solution, only 200 ppm P-2 can protect copper very well with maximum inhibition efficiency as high as 92%.

By using simple phosphorylation, phosphonic acids can be introduced to ethyl cellulose. Ben Youcef et al. [47] reported the method of microcapsulation to encapsulate almond oil as inhibitor by using phosphorylated ethyl cellulose (P-3 in Figure 5). It was found that the P-O^−^ functionalized groups strongly interacted with the metal ions in the metal substrate, and therefore, generated synergetic anti-corrosion effect for almond oil.

Comparing with the post-functionalization of natural polymer, the direct polymerization of phosphorus-containing monomer provides accurate structure control and rational design of polymeric inhibitors [48]. The most commonly used phosphorus-containing methacrylate-base monomers are shown in Figure 5 (M-1 and M-2). For example, Keil et al. [49] reported UV-cured polyester acrylate coatings on zinc and iron as anti-corrosion surface. Moreover, Pebere et al. [50] extended the study on the UV polymerization of M-1 and M-2 to prepare effective corrosion protection. It was found that coatings containing phosphonic acid methacrylate (M-2) showed better anti-corrosive performance than those containing methacrylate phosphonic dimethyl ester (M-1), owing to the improved adsorption of phosphonic acid to metal surface than corresponding ester. Similarly, Ilia et al. [51] copolymerized vinylphosphonic acid (VPA) and dimethyl vinylphosphonate (DMVP) to prepare PVPA-*co*-PDMVP copolymer P-5 (Figure 5) via free radical photopolymerization. Interestingly, they found that the presence of phosphonate groups from DMVP in copolymers was beneficial and a molar ratio VPA:DMVP 4:1 and 3:1 enhanced the anticorrosion for iron surface in comparison with homopolymer of vinylphosphonic acid (P-4). In other words, the coordination between P-OH and metal surface competes with the formation rate of uniform protective layer. While copolymers with VPA:DMVP 4:1 may show higher diffusion coefficient, and therefore, faster formation of protective film than other polymers.

In addition to introducing additional adsorption sites to polymeric inhibitors, low surface energy monomers have also been used in the design of efficient inhibitor. Moratti et al. [52] prepared block copolymer P-6 (Figure 5) via free radical polymerization of heptafluorodecyl methacrylate and (dimethoxyphosphoryl) methyl methacrylate. The copolymers were then immobilized as a monolayer film to the surface of 316L stainless steel by treatment of dilute solutions in trifluoroacetic acid for 30 min followed by rinsing. Owing to the presence of fluorinated block and the presence of adsorption site from the phosphorus block, the resulting polymeric inhibitor exhibited excellent anti-corrosion property and long-term stability.

Champagne et al. [45] reported the nitroxide-mediated polymerization (NMP) to prepared polystyrene-*b*-poly(dimethyl(methacryloyloxy)methyl phosphonate) and corresponding polystyrene-*b* poly(dimethyl(methacryloyloxy)methyl phosphonic acid) (P-7 in Figure 5) and its graft onto polysaccharide chitosan (P-8 in Figure 5). The controlled radical polymerization feature of NMP allowed rational design of the repeat unit and the topology of the resulting polymer. It can be anticipated that the resulting grafted polymers are potential candidate for excellent inhibitor due to the coexistence of chitosan and phosphorus-containing polymers (Table 2).

In addition to the radical-based polymerization, sol-gel method has also been developed for the preparation of phosphorus-containing polymeric inhibitors. Mandler et al. [53] incorporated phenylphosphonic acid (PPA) to a sol-gel film to enhance the corrosion protection of metallic aluminum; however, it was found that such method resulted in aggregation and phase separation. To overcome such problem, Choudhury et al. [54] proposed the design of networked methacrylate- hybrid by copolymerizing 2-(methacryloyloxy)ethyl phosphate (M-3 in Figure 5), containing a polymerizable methacrylate group and functional phosphate group with 3-[(methacryloyloxy)propyl] trimethoxysilane [55]. It was found that the proposed network not only enhanced the binding of the coating to the metal substrate via the acid-base interaction of the P-O− group of the phosphate with the Mn+ of the metal substrate, but also resulted excellent anti-corrosion property.

**Table 2 materials-16-02954-t002:** Inhibition property of typical phosphorus-containing synthetic polymeric inhibitor.

Inhibitor	Metal	Corrosion Medium	Test Method	Highest IE (%)	Reference
P-1	aluminum alloy 3003	pH = 5 acetic acid	EIS	/	[44]
P-2	copper	200 × 10^−3^ g/L NaCl	weight loss, EIS	92	[46]
P-3	mild steel	3.5% NaCl	salt spray test	/	[47]
M-1/M-2	low-carbon steel	0.1 M NaCl	EIS	85	[49]
M-3	mild steel	3.5% NaCl	PDP, EIS		[54]
P-4/P-5	iron coin	3% NaCl	PDP, EIS	83.5	[51]
P-6	316L stainless steel	trifluoroacetic acid	long term stability tests	/	[52]

### 3.2. Sulfur-Containing Synthetic Polymeric Inhibitor

As a classic coordination atom, sulfur can form a stable complex with different metal ions. Generally, there are two kinds of sulfur-containing polymeric inhibitors: (i) polythiophene and (ii) polysulfone.

Normally, intrinsically conducting polymers or conjugated polymers, such as polythiophene, polypyrrole, and polyaniline, have been widely used as protective coating for corrosion protection of steel [56]. Polythiophene was first reported as corrosion protection in 1989 [57]. The facile electropolymerization of thiophene and its derivatives allows the preparation of homogeneous polymer films on the surface of different metals with good electrical properties and chemical stability [58]. Gonzalez-Rodriguez and coworkers [59] compared the anti-corrosion property of poly(3-octyl thiophene) (P3OT) and poly(3-hexylthiophene) (P3HT) (P-9 in Figure 6). The plate they used was a commercially available 1018 carbon steel sheets, and copper wires were welded to the plate, which was used as a reaction platform for electrode deposition. The polymer solution was then deposited on the electrode, evaporated solvent, dried, and annealed to afford the P3OT/P3HT coating. Both polymer films were found to be effective in protecting the substrate from corrosion by decreasing the critical current necessary to passive the substrate, increasing the pitting potential, and broadening the passive interval, and P3HT was found to be more effective due to a much lower number of defects than P3OT films. Interestingly, P3HT gave better protection than P3OT, because of the lower defects in the film of P3HT than that for the P3OT films. Thiophene can also be copolymerized with other monomers to improve the anti-corrosion property. For example, Branzoi et al. [60] investigated the anti-corrosive properties of poly(*N*-methylpyrrole-Tween20/3-methylthiophene) coatings on carbon steel type OLC 45 in 0.5 M H_2_SO_4_ medium. The surfactant Tween 20 was a dopant used in the electropolymerization process, which could improve the anti-corrosive properties by hindering the corrosive sulfate ion penetration. The corrosion rate of PNMPY-TW20/P3MT-coated OLC 45 has been indicated to be ~10 times reduced in comparison with uncoated OL 45, and the corrosion protection efficiency of the coating is above 90%. More importantly, the anti-corrosion property of such coating can be tuned by the condition of electropolymerization, such as electrodeposition current and time, highlighting the potential application of such technique.

Furthermore, the anti-corrosion performance of polythiophene can be improved by blending with other polymers [61]. Meanwhile, blending can also improve the processability and mechanical strength of the material and reduce the cost of expensive conductive polymers [62]. For example, Nicho et al. [63] blended P3OT with polystyrene (PS) and deposited it onto stainless steel sheets using the drop-casting technique, where a solution of the blend is added dropwise to the steel sheet, the solvent is evaporated, then dried and annealed. Subsequently, the room temperature corrosion behavior of the prepared P3OT/PS coated 304 stainless steel was studied under 0.5 M NaCl. According to their study on the temperature effect, it was found that high temperature (e.g., 100 °C) can increase the adhesion degree between coating and substrate, making the coating less porous and defective to give a denser surface, therefore giving it better inhibition performance. Furthermore, they systematically investigated the anti-corrosion property of P3HT, P3HT/PS and P3HT/PMMA (polymethyl methacrylate) blends coatings on A36 steel corrosion protection in 0.5 M H_2_SO_4_ solution. It was found that blends of P3HT with PMMA and PS improved the protection of steel in comparison with native P3HT. P3HT/PMMA blend gave the best protraction to the steel. To enhance the interface interaction between different polymers in the blend, Huang et al. [64] reported the preparation of P3HT/poly(styrene-*co*-hydroxystyrene) blend (P-10 in Figure 6) via the intermolecular hydrogen bonding between thiophene and phenol group. The hydrogen bonding not only improved the miscibility between two polymers, but also enhanced the adhesion force between iron and coating layer. Compared with native P3HT, the inhibition performance of the blend improved and the decreased upon thermal treatment.

In addition to polythiophene-based inhibitor, polysulfone represents another kind of sulfur-containing polymeric inhibitor. For example, the non-toxic amino acid methionine was used as a sulfur-containing corrosion inhibitor for mild steel because of the coexistence of N, O, and S atoms in one molecule. Moreover, the corrosion inhibition performance of methionine, methionine sulfoxide, and methionine sulfone in HCl for mild steel has been studied previously and moderate inhibition efficiency has been achieved. To enhance the adsorption of inhibitor and metal substrate, polysulfone may be potential strategy [65]. Ali and coworker carried out systematical study on the anti-corrosion performance of series of polysulfones. Butler’s cyclopolymerization of diallyl ammonium salts and their copolymerization with SO_2_ was used to synthesize a series of polysulfones with residues of essential amino acid methionine (P-11 and P-12 in Figure 6) [66]. Especially, in the copolymer P-14 (Figure 6), half of the sulfide was oxidized to corresponding sulfone, which greatly improved the water solubility. More importantly, the copolymer P-14 demonstrated superior inhibition of mild steel in 1 M HCl at 60 °C with inhibition efficiency of 99% at concentration of 25 ppm, while corresponding monomer can only achieve inhibition efficiency of 31% at the same concentration, highlighting the contribution of polymer configuration. In another study, they treated polymer P-11 with H_2_O_2_ to generate corresponding sulfone P-12 and sulfoxide P-13 [67]. All of these polymers can achieve inhibition efficiency (IE) up to 87%, even at a very low concentration of 6 ppm in 1 M HCl. It revealed that the sulfoxide (S=O) base sequence was more effective in mitigating mild steel corrosion in comparison with sulfide (S) and sulfone (O=S=O). The copolymerization methodology also allowed the introduction of multiple adsorption groups to one inhibitor. For example, they [67] also synthesized a new tripolymer P-15 (Figure 6) consisting of carboxylate, sulfonate, and phosphonate using the Butler cyclopoymerization technique and copolymerization with sulfur dioxide. They evaluated the performance of P-15 as a corrosion inhibitor for St37 carbon steel. It was found that the as prepared inhibitor demonstrated protection efficiency of 79.5% and 61.1% in HCl and H_2_SO_4_ media at a concentration of 1000 mg/L, respectively. Interestingly, it was found that the addition of KI can greatly enhance the performance to give IE as high as 93.5%, which may due to the synergistic effect of the cooperative co-adsorption of I^−^ on the metal surface. They also synthesized series of poly(bis-zwitterion) (P-16, P-17, and P-18 in Figure 6) with chelating motifs of [NH^+^(CH_2_)_2_NH^+^(CH_2_CO_2_^−^)_2_] via the same strategy. These polymers were found to be very good inhibitors of mild steel corrosion in 1 M HCl. Similarly, the addition of KI (400 ppm) can generate synergistic effect to achieve 98% inhibition of mild steel corrosion for a duration 24 h at 60 °C [68].

Compared with polysulfone, polythioether can bind with metal surface more strongly due to the S-Fe bond. In order to improve the anti-corrosion property of native polythioether, our group designed a series of cobaltocenium-containing polythioether type metallo-polyelectrolytes (P-19, P-20, and P-21 in Figure 6). The synthesized diolefin monomers were first sulfonated to introduce sulfonate groups, followed by photo-induced thiol-ene polymerization to synthesize the polymer, then azidation and copper-catalyzed post-click modification to graft the cobalt dichloride groups to the polysulfide backbone to obtain the target polymer (The synthesis process is shown in Figure 7). It can be found that there are multiple interactions between these polymers and metal surface, such as coordination between S and metal, triazole and metal, electrostatic interactions, and the potential ion-π interaction. According to the weight loss experiments and electrochemical study, all these polymers were found to be effective inhibitors, which can achieve inhibitive efficiency as high as 95% at concentration as low as 10 mg/L. Moreover, our study also revealed the structure–property relationship for the design of new polymeric inhibitor, highlighting the important role of flexible linkage between the polymer main-chain and the charged group and the number of charged groups [69].

In addition to the above sulfur-containing polymeric inhibitors (Table 3), sulfur has also been introduced to resin coating. For example, Mohammad El-Sawy and coworkers compared the inhibition performance of modified urea (P-22) with thiourea formaldehyde (P-23) resins for steel surfaces. As expected, owing to the presence of sulfur atoms in thiourea resin, P-23 demonstrated the best protection performance and adhesion [61].

### 3.3. Nitrogen-Containing Synthetic Polymeric Inhibitor

Nitrogen-containing synthetic polymeric inhibitor represents a major class of polymeric inhibitors. Most of the nitrogen-containing polymeric inhibitors belong to polyelectrolyte, such as poly(quaternary ammonium), polyethyleneimine, polyaniline, and so on.

#### 3.3.1. Poly (Quaternary Ammonium)

Generally, polymeric corrosion inhibitors tend to outperform their monomer counterparts due to advantages, such as increased adsorption sites and entropy processes that displace water molecules from the metal surface. Since quaternary ammonium salts-based small molecular corrosion inhibitors have been widely used in metal corrosion inhibition, their corresponding poly(quaternary ammonium) salts have also been investigated as potential inhibitor owing to their advantages of high water solubility [70].

As discussed in the section of sulfur-containing polymeric inhibitors, polyquaternary ammonium salts are usually synthesized via the Butler’s cyclopolymerization of diallyl ammonium salts. For example, as shown in Figure 8, Ali et al. [66] prepared a variety of unsaturated *N*,*N*-diallyl compounds from 1,6-hexanediamine and performed cycliopolymerization to synthesize a series of water-soluble polyquaternium oligomers (P-24, P-25, and P-26). All these inhibitors demonstrated good corrosion inhibition property with IE as high as 93%.

#### 3.3.2. Polyethyleneimine

In addition to the polyquaternary ammonium salts, the commercially available polyethyleneimine (PEI) and its derivatives have also been broadly used as inhibitor. In principle, the corrosion inhibition efficiency of PEI is related to the number of sub-methyl groups and the number of vinyl and amino groups, which may influence the adhesion and adsorption properties with metal substances. Moreover, the different types of nitrogen atoms in PEI may also be protonated under the condition of acid corrosion, further enhancing the adsorption affinity [66].

Milošev et al. investigated the corrosion protection of PEI (P-27 in Figure 9) for ASTM 420 stainless steel [71] and the relationship between anti-corrosion property and the molecular sizes of PEI [72]. It was found that PEI is a corrosion inhibitor against pitting corrosion, and PEI with an average molar mass of 2000 g/mol shows the best effect against localized corrosion. Moreover, the amine groups in PEI can be protonated by CO_2_ in water, PEI can also be used as green inhibitor under the condition of saturated CO_2_ solution. Umer and coworkers studied the effect of temperature on the corrosion behavior of API X120 steel in a saline solution saturated with CO_2_ in absence and presence of PEI [72]. It was found that PEI significantly decreases the corrosion rate of API X120 steel with inhibition efficiency of 94% at a concentration of 100 μmol/L. This is attributed to the fact that PEI molecules adsorb onto the metal surface through heteroatoms, forming a dense protective film that limits the transfer of aggressive ions to the metal surface and reduces the corrosion rate. While the addition of PEI with saturated CO_2_ works best because PEI adsorbs on the steel surface and has the ability to trap CO_2_ in the brine solution, with the end result being a reduction in fouling and a smoother steel surface.

PEI can also be used to form hybrids with inorganic material, such as grapheme oxide (GO), to further improve the inhibition performance. As shown in Figure 10, Quraishi et al. [73] modified GO with PEI via facile amidation and investigated its corrosion inhibition of carbon steel in a solution of 15% HCl. According to the weight loss experiment, the PEI-GO alone provided a high corrosion inhibition efficiency of 88.2% at a temperature of 65 °C. Moreover, KI can further increase IE to 95.8% due to their synergistic effect.

As mentioned, PEI can be potentially quaternized to generate poly(quaternary ammonium) salt. Gao et al. [74] prepared quaternary polyethyleneimine (Q-PEI, P-28 in Figure 9) via tertiary amination reaction and subsequent quaternization reaction, and investigated the corrosion inhibition performance of Q-PEI on mild steel (A_3_ steel) in H_2_SO_4_ solution. It was found that Q-PEI is a very effective corrosion inhibitor for A_3_ steel, and the corrosion inhibition rate was up to 92% when immersed in 0.5 M H_2_SO_4_ solution for 72 h at a concentration of only 5 mg/L. Another study [75] further investigated the corrosion protection mechanism of PEI and Q-PEI on Q235 carbon steel under different acid mediums. It was found that Q-PEI is more cationic compared with PEI. The chemisorption that occurs between Q-PEI and carbon steel surface is mainly influenced by three aspects: (i) the electrostatic interaction between the positive charge of the amine nitrogen atom and the electronegative carbon steel surface; (ii) the covalent bond formed by the electrons on the phenylbenzene that can enter the three-dimensional orbitals of the iron atom; and (iii) the nitrogen atom of Q-PEI and iron atoms on the surface of carbon steel to form chelate complexes.

From the previous results of Ingrid et al. [71], it can be seen that there is still a possibility to improve the corrosion protection performance of PEI as a corrosion inhibitor for mild steel in neutral media. To further enhance the performance of PEI, Fu et al. [76] modified PEI with phosphorus groups to give polyethyleneimine phosphorous acid (PEPA) (P-29 in Figure 9). Owing to the synergetic effect from nitrogen atoms and the phosphorus groups, P-29 can be used as corrosion inhibitor in neutral medium for mild steel. Subsequently, they investigated the influence of molecular weights of PEPA on the anti-corrosion property [77]. Generally, three kinds of PEI with molecular weight of 600, 1800, and 10,000 were used. The results showed that PEPA is an excellent corrosion inhibitor, and the best inhibition performance was achieved at a dosage of 200 mg/L for PEPA synthesized from PEI with a molecular weight of 1800, with an inhibition rate of 94.6%.

#### 3.3.3. Polyaniline

As a typical conductive polymer, polyaniline (PANI, P-30 in Figure 11) and its derivatives have been widely used as corrosion inhibitors in the form of either solution or coatings. Generally, the mechanism of PANI corrosion protection can be attributed to three effects: (i) the physical corrosion protection due to PANI coating separating the metal from the corrosive medium [78]; (ii) the passivation effect of PANI catalyzing the formation of oxide film on the metal surface [79]; and (iii) the heteroatom in PANI can form some kind of bond with metal atoms and perform a role in corrosion inhibition. As early as 1985, Deberry [80] reported the application of electrochemically synthesized polyaniline coating for corrosion protection of ferritic stainless steel in H_2_SO_4_. It was found that polyaniline could provide anodic protection for stainless steel and significantly reduce the corrosion rate of stainless steel. In a later study, Wessling found that polyaniline coating synthesized from dispersion may lead to significant shift of the corrosion potential and generated so-called passivation of metal. Similarly, Mirmohseni et al. [81] synthesized polyaniline polymers chemically and casted from 1-methyl-2-pyrrolidone (NMP) solution to iron samples. Such coating was found to be an effective inhibitor for iron in various corrosive environments, such as NaCl (3.5 wt%), tap water, and acids. Meanwhile, polyaniline was also found to be a more powerful inhibitor than conventional polymer, such as polyvinyl chloride.

The effect of substituents, such as −OCH_3_, −COOH, and −CH_3_, in the structure of polyaniline on corrosion inhibition has been investigated [82]. For example, Bereket and coworkers prepared poly(*N*-ethylaniline) (P-31 in Figure 11) coatings via cyclic voltammetry method on copper [83].

In addition to the polyaniline and substituted polyaniline (Table 4), Venkatachari et al. [84] synthesized water-soluble poly(*p*-phenylenediamine) (P-32 in Figure 11) and investigated its corrosion inhibition of iron in different concentrations of 1 M HCl using polarization technique and electrochemical impedance spectroscopy. The corrosion inhibition efficiency of poly(*p*-phenylenediamine) was found to be 85% at a concentration of 50 ppm, and even when the concentration of its monomer was scaled up to 5000 ppm, the corrosion inhibition efficiency of the monomer was only 73%. Compared with polyaniline, the corrosion inhibition performance of poly(*p*-phenylenediamine) is also found to be superior. Venkatachari et al. [85] found that poly(diphenylamine) also has good corrosion inhibition properties and the corrosion inhibition efficiency can be as high as 96% even at a very low concentration of 10 ppm. The analysis of FT-IR revealed that poly(diphenylamine) has a strong adsorption effect on the iron surface, which can improve the passivation of iron in sulfuric acid.

Polyaniline can also be copolymerized with other monomers to exert corrosion inhibition. Shi et al. [86] obtained poly(aniline-co-o-anthranilic acid) (P-33 in Figure 11) by copolymerizing polyaniline with anthranilic acid and investigated the corrosion inhibition efficiency of polyaniline copolymer in HCl solution on carbon steel. The results showed that the polyaniline copolymer solution in HCl solution had an effective corrosion inhibitor effect on carbon steel, and the adsorption on the metal surface followed the Langmuir adsorption isotherm with an IE value of 90.5% when the concentration was 20 ppm. Shukla et al. [87] investigated the slow-release properties of a series of anilines-formaldehyde polymers in 1 M HCl solution on mild steel. The three copolymers synthesized, poly(aniline-formaldehyde) (P-34), poly(o-tolylene-formaldehyde) (P-35), and poly(p-chloroaniline-formaldehyde) (P-36), showed excellent corrosion inhibition, and poly(p-chloroaniline-formaldehyde) had the best corrosion inhibition efficiency due to the presence of Cl in the structure and the interaction of the lone electron pair of chlorine atom with the metal surface, with an IE value of 98% at a concentration of 10 ppm.

Seegmiller et al. [88] prepared coatings by blending the conductive polymer polyaniline with the vinyl polymer poly(methyl methacrylate) and investigated the anticorrosive properties of the blended films on different metals in 1 mol/L H2SO4. The as-prepared conductive blends were effective in protecting Cu, Ni, Fe, and other metals [89].

#### 3.3.4. Inorganic Mineral-Doped Polyaniline-Based Inhibitor

The doping of inorganic materials, such as montmorillonite, zeolite, and silica, in polyaniline can enhance the gas barrier properties of the composite [90], which can make the path that corrosive substances, such as oxygen and moisture, pass through the coating more tortuous, making the corrosive medium less corrosive and thus achieving anti-corrosion effects. Wei et al. [91] investigated the effect of alkali and acid doping forms of polyaniline on cold rolled steel using electrochemical corrosion measurements and compared it with undoped polyaniline coatings. They found that the polyaniline-based and epoxy topcoats treated with zinc nitrate had better overall protection compared to other coating systems, which is consistent with the observations of Wessling et al. [79] Yeh et al. [92] prepared polyaniline-clay nanocomposites (PCN) by doping montmorillonite (MMT) clay into polyaniline. It was found that PCN has 400% lower permeability and better corrosion resistance than conventional polyaniline. In a similar study, Olad et al. [93] prepared two kinds of nanocomposites, PANI/Na-MMT and PANI/O-MMT, by doping polyaniline with organophilic montmorillonite (O-MMT) and hydrophilic montmorillonite (Na-MMT). Although the conductivity of the nanocomposite was lower than that of polyaniline films, both hydrophilic and organophilic PANI/MMT nanocomposite coatings outperformed the pure polyaniline coating for corrosion protection of iron samples.

Alloys, such as stainless steel and magnesium alloys, can also be protected from corrosion with PANI/MMT nanocomposite coatings. The release of metal ions, such as iron, chromium, and nickel from the biological environment surrounding the alloy leads to reduced biocompatibility and susceptibility to local attack, which limits its use in biomedical applications [94]. Stainless steel is protected by a passive film formed on the surface, and the most serious problem in practical application is localized corrosion [95]. Mehdi et al. [96] used direct electrochemical galvanostatic method to electrochemically synthesize polyaniline and PANI/MMT nanocomposite coatings on the surface of 316L stainless steel, and the studied the corrosion protection performance in 0.5 M HCl medium. It was found that the inhibition efficiency was up to 99.8%, which indicated that the coatings have outstanding potential in protecting 316 L SS against corrosion in acidic media. Shao et al. [97] investigated the corrosion protection of AZ91D magnesium alloy by epoxy coating containing PANI/OMMT powder in 3.5% NaCl solution. It was found that the lamellar structure of PANI/OMMT could improve the barrier ability of the coating to electrolyte solution and PANI could form an oxide layer to enhance the corrosion resistance of magnesium alloy. Even after 6000 h of immersion, PANI/OMMT coating could still maintain the good corrosion resistance of AZ91D magnesium alloy.

In addition to montmorillonite, carbon-based inorganic materials, such as graphene, carbon nanotubes, and glass fibers (GB), can also be doped with polyaniline to afford composites for corrosion protection. Conductive graphene has also attracted interest in recent years due to its high aspect ratio of about 500 [98], and the combination of graphene with polymers has also been found to have good mechanical [99] and galvanic properties [100], and potential barrier properties [101]. Yeh et al. [102] doped graphene with polyaniline to obtain polyaniline/graphene composites (PAGCs). In comparison with pure polyaniline, the hybrid PAGCs were found to have excellent barrier properties to O_2_ and H_2_O.

The good electronic conductivity of carbon nanotube endows the resulting hybrid excellent corrosion protection performance. Zhu et al. [103] prepared dendrite-like polyaniline/carbon nanotube (PANI/CNT) nanocomposites via the interfacial π-π interactions between carbon nanotubes and polyaniline. The PANI/CNT nanocomposites have good redox ability in acidic and neutral media with excellent physical barrier effect of water and oxygen and metal passivation catalytic effect. Hou et al. [104] prepared polyaniline/glass fiber (PANI/GB, PGB) composites by in situ oxidative polymerization and used these materials to modify epoxy coatings and investigated the effect on their corrosion resistance. The results showed that the addition of PGB composites led to a significant increase in the corrosion resistance of the coatings, and the coatings had the best corrosion resistance when the PANI to GB mass ratio was 1:1.

#### 3.3.5. Polyaniline-Based Inhibitors from Doping with Protonic Acids

The doping of polyaniline with different mineral and organic acids affects the degree of protonation of polyaniline, which affects the anions in the structure of polyaniline, and both the type and concentration of the doped acid affect the conductivity of polyaniline conductive form emeraldine salt (P-37 in Figure 11) [105]. Kohl et al. [106] prepared five polyaniline salts (PANI-HA) using five acids, namely, phosphoric acid (H_3_PO_4_), sulfuric acid (H_2_SO_4_), hydrochloric acid (HCl), *p*-toluenesulfonic acid, and 5-sulfosalicylic acid. The results showed that the type of PANI dopants and pigment volume concentration (PVC) perform an important role in the corrosion resistance of the coating, and the PVC corresponding to the best corrosion inhibition effect is different for different PANI-HA. The general rule of PVC effect on the corrosion resistance of the coating is at low PVC values (PVC = 0.1–5%), the corrosion resistance is independent of the pigment used; when the PVC increases up to 10%, the corrosion resistance becomes significantly worse with the increase in PVC. Generally, the incorporation of PANI pigments into epoxy resins often resulted in severe agglomeration and required sonication [107]. Sadegh et al. [108] prepared glycol-modified epoxy coatings containing polyaniline-camphorsulfonic acid particles on mild steel surfaces using a novel one-pot method without further dispersion of the particles. Zhang et al. [91] doped two acids, hydrofluoric acid and camphorsulfonic acid, as doping acids into PANI subsequently added to epoxy coatings. The EIS results showed that both acids of doped PANI had an improved effect on the retardation performance of epoxy coating, especially the anticorrosion effect of camphorsulfonic acid doped PANI coatings was greatly enhanced. The anticorrosion mechanism of different forms of PANI coatings can be summarized as following: (i) through barrier properties; (ii) through the formation of passivation layer by redox reaction; and (iii) through the formation of insoluble counter anionic salts. Ali et al. [109] prepared self-assembled polyaniline nanotubes using sodium dodecylbenzene sulfonate as the dopant acid and compared their corrosion performance with that of conventional polyaniline. It was found that the corrosion currents of iron sheets coated with polyaniline nanotubes were much lower than those of iron sheets coated with conventional polyaniline. Grgur et al. [110] conducted a more practical study on benzoate doped polyaniline coatings, which were tested for cathodic protection under 3% NaCl, atmospheric and Sahara desert conditions. The results show that the coating can effectively protect mild steel even in a short period of time, and they propose a “switching zone mechanism” (Figure 12).

#### 3.3.6. Polyaniline-Based Inhibitor from Doping with Metals Oxides

In addition to the inorganic substances and protonic acids, metals and their oxides can also be doped with polyaniline to improve the anodic protection of polyaniline, owing to their sacrificial oxidation. The most commonly used metal materials are iron, zinc, titanium, and their corresponding oxides. Venkatachari et al. [111] prepared polyaniline-Fe_2_O_3_ composites with aniline to Fe_2_O_3_ ratios of 2:1, 1:1, and 1:2, respectively, using chemical oxidative method in phosphoric acid medium with ammonium persulfate as the oxidizing agent, combined the composites with an acrylic binder to prepare a primer coated on steel specimens. They found that the best corrosion resistance to 3% NaCl solution of the composite coating was achieved at a ratio of 1:1 of aniline to Fe_2_O_3_. The good anti-corrosion effect of polyaniline-Fe_2_O_3_ composites can be mainly attributed to the barrier effect due to the formation of a passive film and iron-phosphate salt film on the iron surface. The size of the filler particle size in polymer composites has a great influence on the performance of the composite [112], and when the size of the filler particles is reduced to the nanoscale, the performance is usually very different from that of the micron size [113]. It was found that the doping of zinc also effectively improved the electrical conductivity and corrosion resistance of PANI [114]. The highest electrical conductivity of PANI/Zn nanocomposites was obtained when the zinc content was 4 wt%, and the performance exhibited by PANI/Zn nanocomposite films was more excellent than the composite films. Radhakrishnan et al. [115] synthesized polyaniline/nano-TiO_2_ composites using in situ polymerization. In comparison with conventional polyaniline, the addition of nanosized TiO_2_ improved the corrosion resistance of polyaniline coatings, and the corrosion resistance of polyaniline prepared by 4.18% nano-TiO_2_ could be increased to more than 100 times. Al-Masoud et al. [116] prepared bimetallic oxide nanocomposite functionalized with polyaniline, by combining ZnO and TiO_2_ simultaneously with polyaniline (PANI). It was found that the as prepared ZnTiO@PANI is a strong acidic corrosion inhibitor, and its corrosion inhibition rate can reach 98.9% at a concentration of 100 ppm in acidic chloride solution (1.0 M HCl + 3.5% NaCl).

### 3.4. Other Type of Polymeric Inhibitors

In addition to the above well-studied systems, other kinds of polymeric inhibitors have also been investigated because modern polymer chemistry allows the introduction of hydrophilic and adsorption groups to the side chains on the basis of the main chain structure and copolymerization with other monomers (Table 5).

Polyacrylic acid (PAA) is the most well-known vinyl polymer corrosion inhibitor used in the early study of anti-corrosion. Recently, polyacrylate or acrylamide copolymer corrosion inhibitors have become more popular. Lin et al. [117] prepared poly(methyl acrylate)-*co*-poly(acrylic acid imidazoline) (MA-ACI, P-38 in Figure 13) from methyl acrylate and acrylic imidazoline with azo diisobutyronitrile as initiator. According to the rotating hanging plate method, P-38 showed inhibition efficiency as high as 82% at a concentration of 0.10 g/L in 1 mol/L H_2_SO_4_ at 30 °C. Taghi et al. [118] prepared neodymium-poly acrylic acid complex (Nd-PAA, P-39 in Figure 13) by adding neodymium to PAA and applied them to the corrosion protection of ST-12 type in 0.1 M NaCl. Due to the anionic nature of PAA, they deposited densely and crack-free ultrafine Nd-PAA films on the steel surface.

Moreover, chemical modification or grafting of inorganic material by synthetic polymer has also been developed to afford efficient inhibitors. For example, as shown in Figure 14, Yu et al. [119] modified graphene oxide with polystyrene by in situ microemulsion polymerization. The resulted hybrid demonstrated significant improvement in the corrosion resistance in comparison with native graphene and polystyrene, with the corrosion resistance efficiency increasing from 37.9% to 99.5%.

**Table 5 materials-16-02954-t005:** Inhibition property of other types of polymeric inhibitor.

Inhibitor	Metal	Corrosion Medium	Test Method	Highest IE (%)	Reference
P-38	N80 steel sheet	1 M H_2_SO_4_	PDP, EIS	90.2	[117]
P-39	ST-12 type steel sheets	0.1 M NaCl	PDP, EIS	/	[118]
P-40/P-41/P-42/P-43	Mild steel	4 M HCl	weight loss, EIS, PDP	98	[120]

Since water solubility is a crucial problem in the design and application of inhibitors, water soluble polymers, such as waterborne polyurethane, have also been used as inhibitors. Our group developed a series of cobaltocenium-containing polyurethanes (P-40, P-41, P-42, and P-43 in Figure 13) via the reaction between hydroxyl-terminated cobaltocenium monomer and different kinds of commercially available diisocyanates (the synthetic process is shown in Figure 15). The presence of charged cobaltocenium group and its counterion endow the resulting polyurethane good water solubility. Owing to the multiple interactions between these waterborne polyurethanes and metal surface and the inter/intramolecular hydrogen bonding between urethane groups, these polymers can strongly adsorb on the metal surface and therefore demonstrated excellent corrosion protection property. According to the weight loss experiment, the inhibition efficiency can achieve as high as 98.0% at a concentration as low as 20 mg/mL toward mild steel in 4 M HCl [120].

## 4. Conclusions

Development and application of inhibitors can not only prolong the lifetime of metal materials, but also provide a possible solution to the energy and economic loss caused by metal corrosion. The development of polymer chemistry provides versatile tools to the rational design and facile syntheses of powerful polymeric inhibitors. As discussed in this review, there are several advantages of polymeric inhibitors:

(i) First, to increase the inhibition efficiency, different adsorption groups with heteroatoms, such as P, S, N, and O, have been introduced to the skeleton of polymer;

(ii) Second, the supramolecular structure of polymeric inhibitors also provides environment adapted configuration of adsorption groups on the metal surface, which enables the synergetic anti-corrosion performance;

(iii) Third, the soft matter nature of polymers enables their integration with different kinds of micro/nano fillers, which can further enhance the anti-corrosion property and therefore improve the actual performance of corrosion inhibitor.

Though a flourishing future of polymeric inhibitor could be envisioned, there are several challenges that limit the further development of polymeric inhibitor and its practical applications. First, a systematic study on the structure-property relationship, especially the relationship between adsorption properties and the chemical structure of inhibitor, should be established as a guide for the design of high efficient inhibitor. Different from small molecular weight inhibitors, the influence of the structure of main-chain and side groups and the molecular weight (number of repeat unit) should be well studied. More importantly, although quantum chemistry, such as density functional theory density, has been used to understand the process of anti-corrosion, artificial intelligence, and computer-aided structural design should be developed to the design of polymeric inhibitors. Second, since most of the polymeric inhibitor has been used as anti-corrosion coatings, the integration of polymeric inhibitors and inorganic fillers and the method to prepare composites should also be focused. Moreover, suitable in-situ characterization of the corrosion process and the structure characterization for the change of inhibitor and the metal surface should be developed. Third, since the broad application of inhibitor has already caused environment pollution, green inhibitors, such as polymeric inhibitors from natural sources, should be investigated. While for the synthetic polymeric inhibitor, facile synthetic methodology and economic issue should be considered. Fourth, with regarding to the practical application of inhibitor in the form of coating, the integration of stimulus-responsive property, such as self-healing, with polymeric inhibitor may be a further direction for the development of smart inhibitors.

## Data Availability

The data presented in this study are available on request from the corresponding author.

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
