# Peer review of "Recent Progress of Polymeric Corrosion Inhibitor: Structure and Application"

_materials, 2023, doi:10.3390/ma16082954_

Round 1

Reviewer 1 Report

1- Figure 10 need to be replaced as its quality is very law. 

Author Response

Q1. Figure 10 need to be replaced as its quality is very law. A: A high-resolution image has been updated according to your suggestion.

Reviewer 2 Report

This manuscript was well prepared. My comments are follows:

1. There are low information related to polymeric corrosion informations, if more informations such as molecular structures, corrosion type, electrolytic medium, and so on, will be tabluated in Tables, it is good for readers.

2. Corrosion inhibition mechanisms needs to describe.

3. Synthesis strategies woulde be described

4. Future suggestions needs to idendify

Reviewer 3 Report

-The author have just gathered some articles conclusion about polymeric inhibitor. There is no information about application, synthesis method, mechanism.

-The title does not match the literature

-In the review process is very important to have structure, there is no structure

-I see no information why this review paper is written because the structure is rambling

-There is no table for any comparison

-All the information in this article can easily be found by a simple search

All in all, writing a review paper is not just gathering some information and stick them together

Therefore, I strongly resist against publication of this paper

Reviewer 4 Report

The manuscript "Recent progress of polymeric corrosion inhibitor: design, synthesis and applications" has good potential in scientific terms and high practical significance. The analysis carried out is worthy of publication.

However, the following remarks to the manuscript:

- In the introduction on lines 113-117, the sentences are unrelated. The first sentence refers to "corrosion inhibitors based on natural polymers", while the second sentence refers to "corrosion inhibitors based on synthetic polymers";

- Supplement Scheme 1 with a few examples of polymeric inhibitor films, if possible, in order to respect copyright;

- In lines 147-148, the abbreviation "AA" is introduced, but it doesn`t appear further in the text;

- On line 283, the abbreviation "IE" is not deciphered;

- For Figures with chemical formulas, add links to primary sources (for example, a chemical reference book, etc.);

- On line 197, delete the phrase "by means of physical blending";

- Supplement the proposal on lines 248-249 with information on the method and equipment for testing steel samples;

- Sulfur has a good affinity for iron...so in the sentence on lines 302-303 correct the phrase "sulfur has also been introduced to resin coating owing to its good affinity to metal surface";

- On line 436, the "" character is missing;

- In lines 447, the abbreviation "CRS" is introduced, but it doesn`t appear further in the text;

- In the sentence on line 455, remove the gap;

- In lines 504-505, the abbreviations "PTSA" and "SSA" are introduced, but they do not appear further in the text;

- The sentence in lines 544-547 is too complex due to the repetition of its individual phrases;

- The sentence on lines 547-548 is poorly connected with the previous text;

- Check if Celsius degree symbols are correct (lines 242, 269, 289, 359, 574). An example of a more similar character is on line 83, except that there is no space;

- References to literary sources should be formatted in accordance with the requirements of the journal, double-check the formatting guidelines for authors.

Round 2

Reviewer 2 Report

My comments are answerred

Author Response

revised manuscript was updated